# Global Stress Detection Framework Combining a Reduced Set of HRV Features and Random Forest Model

**DOI:** 10.3390/s23115220

**Published:** 2023-05-31

**Authors:** Kamana Dahal, Brian Bogue-Jimenez, Ana Doblas

**Affiliations:** Department of Electrical and Computer Engineering, The University of Memphis, Memphis, TN 38152, USA; kdahal1@memphis.edu (K.D.); bbgjmnez@memphis.edu (B.B.-J.)

**Keywords:** stress detection, wearable device, ECG, HRV features, feature selection, global training, individual testing, machine learning

## Abstract

Approximately 65% of the worldwide adult population has experienced stress, affecting their daily routine at least once in the past year. Stress becomes harmful when it occurs for too long or is continuous (i.e., chronic), interfering with our performance, attention, and concentration. Chronic high stress contributes to major health issues such as heart disease, high blood pressure, diabetes, depression, and anxiety. Several researchers have focused on detecting stress through combining many features with machine/deep learning models. Despite these efforts, our community has not agreed on the number of features to identify stress conditions using wearable devices. In addition, most of the reported studies have been focused on person-specific training and testing. Thanks to our community’s broad acceptance of wearable wristband devices, this work investigates a global stress detection model combining eight HRV features with a random forest (RF) algorithm. Whereas the model’s performance is evaluated for each individual, the training of the RF model contains instances of all subjects (i.e., global training). We have validated the proposed global stress model using two open-access databases (the WESAD and SWELL databases) and their combination. The eight HRV features with the highest classifying power are selected using the minimum redundancy maximum relevance (mRMR) method, reducing the training time of the global stress platform. The proposed global stress monitoring model identifies person-specific stress events with an accuracy higher than 99% after a global training framework. Future work should be focused on testing this global stress monitoring framework in real-world applications.

## 1. Introduction

According to the World Health Organization, stress is any form of change that causes physical, emotional, or psychological pressure [1]. Stress can lead to emotional and mental symptoms such as fear, anxiety, sadness, panic attacks, and depression. Additionally, stress can lead to physical symptoms, including elevated heart rate, difficulty breathing, disruption in sleeping patterns, change in eating habits, difficulty concentrating, and worsening of pre-existing health conditions. People under stress may increase the use of alcohol, tobacco, and other drugs, showing a strong dependence between stress and addiction to drugs and alcohol [2]. To some extent, everyone endures stress in their daily life. In particular, Americans are one of the most stressed-out populations in the world, experiencing a stress level 20 percentage points higher than the global average [3]. Whereas small amounts of stress for short periods can lead to positive changes in our lives, stress becomes harmful when it occurs for too long or is continuous (i.e., chronic). Chronic stress increases the risk of developing major health conditions and diseases such as diabetes, depression, heart disease, and cancer [4]. Monitoring an individual’s stress level regularly can help us identify high-stress situations, enabling us to implement early stress interventions for better management. Effective stress management would reduce the probability of developing major health conditions, improving the well-being of individuals in our society. A direct consequence of improved stress management is the associated reduction in the individual and national economic cost of healthcare, increasing the economic competitiveness of any country.

Accurate stress monitoring can be provided through measuring cortisol levels using blood, urine, or saliva fluids. These methods require a specific laboratory or hospital to analyze the sample and determine the cortisol level. These methods, which can be considered minimally invasive, provide a brief glimpse of cortisol levels at a particular moment, restricting their use to continuous long-term stress studies due to their cost and the inconvenience of requiring an external facility to provide a quantitative measurement. Traditional stress detection methods are based on questionnaires in which subjects answer a validated set of stress-related questions to assess their stress level. Nonetheless, these questionnaires cannot be used for instantaneous stress detection. Real-time stress detection methods have been accomplished through analyzing imaging-based approaches that monitor any changes in individual’s facial expressions, including changes in the blinking rate, pupils, and eyebrows [5,6]. However, these methods could be unreliable due to participants’ manipulated answers and facial expressions regarding mental stress.

Over the last decade, our research community has investigated physiological signals measured using various sensors or wearable devices to identify and measure stress levels. Blood volume pressure, electrocardiography (ECG), electromyography (EMG), electrodermal activity (EDA), and heart rate variability (HRV) are the most used physiological signals to identify stress. Reported research has used these signals independently or in combination with each other to improve the accuracy of the stress detection method. Current research in stress detection methods involves the combination of multiple physiological signals with machine learning (ML) or deep learning algorithms [7,8,9,10,11,12,13,14,15,16,17,18,19,20,21,22,23,24,25,26,27,28,29,30].

Although multiple physiological signals have been investigated and reported in the literature, HRV signal is currently one of the most used signals for assessing mental stress [8,9,10,11,12,13,14,18,19,20,21,22,23,26,27,30] because of its accessibility in current wearable devices. For example, most wristband devices have a photoplethysmography (PPG) sensor to measure HRV features. This paper focuses on stress detection through combining HRV features with ML algorithms to further investigate a low-cost, simple, and accurate stress detection model. Several research groups have worked on this goal. For example, in 2015, Munla et al. [9] investigated a stress detection method in people driving using HRV-based features and ML models. Twenty-two HRV features were obtained from an ECG sensor: maximumRRI, minimumRRI, medianRRI, meanRRI, meanHR, stHR, SDNN, RMSSD, NN50, pNN50, SDNNi, SDANN, TINN, VLF, HF, LF, LF/HF, SD1, and SD2. Among the different ML models, the authors investigated standard classifiers such as vector machine (SVM), K nearest neighbor (KNN), and radial basis function (RBF). Using the 22 HRV features, the authors reported the highest accuracy of 83.33% for an SVM classifier model to detect two physiological states of drivers (i.e., highly stressed, and normal). In 2020, Nkurikiyeyezu et al. proposed person-specific and generic models using a random forest (RF) model, and hybrid-calibrated the model using extremely randomized trees (ExtraTrees) [22]. In that work, the authors extracted HRV- and EDA-based features from the WESAD and SWELL datasets. In case of HRV features, the authors used 75 and 40 HRV features from SWELL and WESAD, respectively, including RMSSD, SSDD, SDRR_RMSSD, pNN25, pNN50, SD1, SD2, RELATIVE_RR, VLF, LF, HF, and LF/HF. Using these HRV-based features, the accuracy of their best RF model was 95.2 ± 0.5% for a person-specific model and 42.5 ± 19.9% for a generic model using the SWELL dataset. The accuracy of the proposed RF model improved for the WESAD dataset, being 98.9 ± 2.4% for a person-specific model and 83.9 ± 13.2% for a generic model. The performance of the hybrid model was trained on the SWELL datasets and achieved 93.09% accuracy with 100 calibration samples per subject. More recently, in 2021, Dalmeida et al. developed a predictive model to accurately classify stress levels using five different ML methods such as KNN, SVM, multilayer perceptron (MLP), RF, and gradient boosting (GB) from ECG-derived HRV features collected from wearable devices [26]. The authors used HRV metrics from a public dataset, DriverDB, as well as created a custom dataset through collecting HRV measurements from four Apple watch users. Using Pearson’s correlation, recursive feature elimination (RFE), and ExtraTrees classifier, they selected the best 11 HRV features that were relevant in the context of the smartwatch: HR, RESP, TP, interval in seconds, AVNN, SDNN, RMSSD, pNN50, HF, LF, LF_HF. The authors reported the MLP model as the best stress-predictive model on the DriverDB dataset, providing a 75% area under receiver operator characteristic curve (AUROC), 80% recall, and 72% F1 score. The best MLP model was then tested on the four Apple Watch users, being able to predict relaxing and stress conditions with 79% and 71% accuracy, respectively. In that same year, Szakonyi et al. proposed an efficient method for an acute stress detection model using HRV features from ECG signal [27]. The authors created a custom dataset from seven participants using portable ECG sensors to record inter-beat (RR) interval data. Using Kubios HRV software [31,32], the authors extracted 52 HRV features from RR interval data. The 52 HRV-based features were classified as 13 time-domain features, 32 frequency-domain features, and 7 non-linear features. Among the different ML algorithms tested, the authors reported the best stress-predictive model using only time-domain and non-linear HRV features (20 features = 13 + 7) and an XGBoost tree algorithm, yielding 96.31% accuracy, 95.23% sensitivity, 97.38% specificity, and 96.26% F1 score. Whereas the performance of the model was not significantly different if all HRV features were used (i.e., 96% accuracy, 95.69% sensitivity, 96.31% specificity, and 95.99% F1 score), the authors identified that the frequency-domain HRV-based features were the ones leading to the lowest predictive model with 92.10% accuracy, 90.81% sensitivity, 93.39% specificity, and 91.96% F1 score. This year, Benchekroun et al. proposed stress detection models based on HRV features using ML algorithms such as logistic regression (LR) and RF [30]. They extracted 61 HRV features using the pyHRV and TSFEL toolboxes from the Multi-Modal Stress Dataset (MMSD) and the University of Waterloo stress dataset (UWSD). Whereas the trained LR model using leave-one-subject-out cross-validation provided a specificity of 66% for the MMSD, the specificity of an RF model was 70% for the UWSD dataset. The authors also investigated a cross-dataset analysis through training on the MMSD dataset and testing on the UWSD one, achieving an RF model with a precision of 63%.

Despite all efforts reported in the literature, our community has not agreed yet on the number of features to identify stress conditions using wearable devices. The number of used features ranges from 7 to 75 among the reported works in the literature. In addition, most works reported in the literature are focused on individual training and testing. Stress varies from person to person; therefore, a stress detection model based on one subject might not provide accurate stress detection with another subject. An effective stress detection model should provide accurate stress classification for any individual in our society. Thanks to our community’s broad acceptance of wearable devices (i.e., people wear wearable devices during their daily activities), this work investigates a hybrid global stress monitoring framework with generic training and person-specific testing. In other words, the model is trained using data from multiple subjects, but its performance is tested individually on each subject. A similar approach was used by Nkurikiyeyezu et al. in their hybrid-calibrated model using extremely randomized trees (ExtraTrees) [22]. Nonetheless, in that work, the authors used HRV- and EDA-based features. Our hybrid framework enables the model to learn general patterns across multiple subjects via combining HRV-based features with an ML algorithm. Among the different HRV features, our global stress model only uses eight HRV features selected using the minimum redundancy maximum relevance (mRMR) method [33]. The mRMR method has been previously reported in the literature to select features using HRV features [18,19], selecting a subset of features with the highest correlation with the target variable but the lowest correlation among themselves [33]. We have validated the proposed global stress model using two open-access databases (the WESAD and SWELL databases). A key feature of the proposed framework is that the penalization on the person-specific stress prediction is negligible, identifying individual stress events with an accuracy higher than 99% after a global training framework. Although this pilot model should be further investigated, we believe this study provides a stepping-stone to developing a global stress monitoring and management model for real-world applications.

The rest of the paper is organized as follows: Section 2 describes the methodology used in this study, describing the proposed global stress detection framework. Section 3 discusses the results obtained using the SWELL and WESAD datasets as well as their combination. Finally, Section 4 summarizes the main achievements of this work and discusses future work to validate the proposed global stress framework further.

## 2. Methods

Figure 1 shows our proposed global stress detection framework that takes HRV-based features generated from individual wearable devices. All users’ data are saved in the cloud for global training. The cloud stores all users’ data, combines them, and trains a single integrated model using an RF algorithm. This remotely trained model is then used for individual stress detection locally. The only requirement of the proposed framework is that the training should occur on a cloud, requiring data transfer from the device to a cloud. However, most wearable devices such as the Empatica E4 and Fitbit devices already have this transfer implemented, making its deployment feasible for real-world applications. The proposed framework is further described in the flowchart, including each step involved in the methodology of the proposed system, shown in Figure 2.

### 2.1. SWELL and WESAD Datasets

We have used two benchmark datasets: the SWELL and WESAD datasets. We used only chest sensors’ data from these datasets. The WESAD dataset is a multimodal dataset containing physiological and motion data recorded using a wrist- and chest-worn device from 15 subjects during a laboratory study [34]. The datasets include physiological signals such as three-axis acceleration (ACC), ECG, body temperature (TEMP), respiration (RESP), EDA, and EMG. During the laboratory study, each subject was exposed to three different physiological conditions: amusement, baseline, and stress. The labeling of these classes was based on the following three events. For example, data were labeled as baseline (i.e., class 0) when the participants were reading magazines for 20 min in a standing/sitting position. During the amusement condition (i.e., class 1), the subject watched eleven funny video clips. Finally, a stress condition (i.e., class 2) was induced when the participants took the Trier Social Stress Test (TSST) [35], which consists of public speaking and mental arithmetic operations.

The second dataset is the SWELL dataset, which contains data from 25 subjects during regular office hours [36]. This multimodal dataset contains information regarding computer logging, facial expression, bodily postures, ECG signals, and skin conductance (i.e., EDA). The SWELL dataset is classified into three classes: no stress, interruption, and time pressure. The labeling of the ‘no stress’ class (i.e., class 0) is related to the participant’s performance in assigned tasks for a maximum of 45 min without time pressure or interruption. The SWELL dataset includes two stress-related events due to interruption (i.e., class 1) and time pressure (i.e., class 2). For example, the workers received interrupting emails during their assigned tasks to interrupt their activity in class 1. Whereas some emails did not require immediate action from the participants (e.g., informative emails), other emails required immediate action by the participants. Finally, in the time pressure condition, the execution time to complete assigned tasks was reduced by two-thirds of the no-stress execution time.

### 2.2. Proposed Global Stress Model

As previously mentioned, we have only used HRV features in this work. The main reasons for only using HRV features are the reduction of the computational tasks and the required sensors, leading to a low-cost stress monitoring device. The HRV features can be extracted from ECG signals. The process of extracting HRV features from ECG signals is found in Ref. [22] using a methodology proposed by the European Society of Cardiology Task Force [37]. The preprocessed SWELL and WESAD datasets containing only HRV features can be downloaded from Kaggle [38]. The preprocessing procedure is described in Ref. [22]. The overview of these datasets is explained in Table 1. For binary classification (stress versus no-stress conditions), the baseline and amusement classes in the WESAD dataset were combined and formed a new class named no-stress. On the other hand, the time-pressure and interruption stress classes in the SWELL dataset were combined into a new class named stress. Finally, we have combined both binary datasets for the global stress platform.

As Figure 2 shows, the dataset was divided into training and testing datasets using a ratio of 70/30 (training/testing). In other words, the training dataset contains 70% of data instances of each subject dataset, and the testing dataset has 30% of data instances. Data standardization is related to the scaling of the features so that their range is the same, ensuring that no features have a higher impact on the model. Whereas data standardization is commonly implemented since some ML algorithms are sensitive to the scale of the input features, we have found that data standardization did not improve the model’s performance in our global stress model (Figure 2). Therefore, we have decided to not implement it, reducing the data processing steps.

### 2.3. Feature Selection

Training all features is impractical due to computational and storage constraints in the proposed global framework. Therefore, selecting the best features (i.e., the feature with the highest classifying power) would reduce the computational complexity of the model, decreasing the training time. In this study, we have selected features based on the mRMR method [33]. Several works reported in the literature have used the mRMR method to select features using HRV features [18,19]. This mRMR approach chooses a subset of features with the highest correlation with the target variable but the lowest correlation among themselves [33]. The features are chosen one at a time using a greedy search method based on optimizing an objective function that balances relevance and redundancy. For our global stress monitory framework, we chose the top eight features selected using the mRMR approach to train and test our model.

### 2.4. Random Forest Algorithm

Among the different ML models, the RF algorithm is one of the most effective for detecting stress due to its better results as compared to others [23]. Some articles that successfully used RF for detecting stress are Refs. [11,15,17,22,26,27]. The RF model is an ensemble of several decision tree algorithms. The objective of this RF model is finding a subset of features randomly among all features to be the base of constructing each decision tree [39]. As a result, each decision tree classifies any new dataset or unknown instances, which are categorized into a specific class based on the majority of the decision trees. Figure 3 shows the structure of an RF classifier, highlighting that the output of the RF classifier is the popular vote among the outputs of the individual decision trees. The hyperparameters of the proposed RF model were determined using grid search, identifying the values that yield the highest performance of the RF model. The best hyperparameter values were max_depth = 20, min_samples_split = 2, and n_estimators = 100. The hyperparameter of the maximum depth was found through analyzing the model’s accuracy for the three datasets (see Appendix A). Because the accuracy of the model remains almost invariant from a maximum depth of 20 to 60, we have chosen 20 to reduce the complexity of the model.

Regarding the validation method of the RF model, we have implemented the stratified K-fold cross-validation method on the 70% training dataset. The stratified K-Fold is preferred over K-Fold for classification problems with unbalanced class distributions [40] since it maintains the proportion of samples for each class in each fold, ensuring each fold has a similar class distribution as the entire dataset. In other words, the stratified K-fold cross-validation method enables the model to be trained on a representative sample of the minority class. Although K = 10 is the preferred value, we used K = 15 since the higher the K value, the lower the prediction error as the model sees more available data [41].

We used Google Collab with Python 3 to investigate the proposed global stress framework. The code was implemented using Python libraries such as TensorFlow, Matplotlib, NumPy, Pandas, and Karas. The computational specs for the algorithm are 12 GB RAM using Google Collaboratory. The code is available in our GitHub [42].

### 2.5. Evaluation Metrics

The evaluation of the trained RF model is done through estimating the confusion matrix (Table 2), which classifies the data instances into positive and negative categories. For example, true positive (TP) is the number of instances when the classifier correctly identifies stress events. False positive (FP) is the number of instances that the classifier identifies true stress events that are not. False negative (FN) is the number of instances in which the classifier labels true stress events as no-stress events, failing to detect stress. Finally, true negative (TN) is the number of instances in which the classifier correctly detects no-stress events. Based on these values, the model’s performance is assessed through measuring the accuracy, precision, recall, and F1 score. The accuracy provides a score of the proportion of correct predictions out of all predictions made. The precision value provides a measurement of true positive predictions with respect to all positive predictions. The recall value provides the ratio of true positive predictions out of all actual positive cases. Finally, the F1-score is the harmonic mean between the precision and recall scores. The value of all these performance metrics ranges between 0 and 1, with 1 indicating the highest performance of the classifier model. These performance metrics are calculated using the following equations:(1)$$\mathrm{Accuracy}=\frac{\mathrm{TP}+\mathrm{TN}}{\mathrm{TP}+\mathrm{FP}+\mathrm{FN}+\mathrm{TN}},$$
(2)$$\mathrm{Precision}\left(P\right)=\frac{\mathrm{TP}}{\mathrm{TP}+\mathrm{FP}},$$
(3)$$\mathrm{Recall}\left(R\right)=\frac{\mathrm{TP}}{\mathrm{TP}+\mathrm{FN}},$$
and
(4)$$F1-score=2\frac{P\times R}{P+R}.$$

## 3. Results

This section explains the features selected and the results obtained from each dataset’s global training (e.g., combining the data from all subjects) and individual testing. The performance of the best-trained model is evaluated on each subject through estimating the accuracy, precision, recall, and F1-score metric parameters.

Firstly, we investigate the SWELL and WESAD datasets separately. The SWELL dataset originally contains 67 HRV features. Applying the mRMR method, we have reduced the 67 HRV features to 8 features: HR_SQRT, pNN25, samples, MEAN_RR, HR, MEAN_RR_SQRT, MEAN_RR_LOG, and MEDIAN_RR. Information on these features is shown in Table 3. These features have enough discriminant power to differentiate between non-stressed and stress events, providing a *p*-value < 0.05 in Tukey’s Honest Significant Differences (HSD) test. These eight HRV features were used as training features of our RF model for detecting stress conditions. We have selected the top eight features because the accuracy of the prediction model is high for all subjects. Whereas the reduction of the number of features penalizes the model’s performance, the increase in them raises the training time. Empirically, we found that eight features are the minimum number of features to discriminate stress events with a high prediction performance and reduced training time.

Figure 4 shows the performance of the trained RF model for each subject of the SWELL dataset. These results are also reported in Appendix A. The classifying power of our RF model is equal to or higher than 99% for all subjects of the SWELL dataset. In other words, the RF model can identify stress events with accuracy, precision, recall, and an F1 score ranging from 98 to 100%.

Next, we train and evaluate the performance of an RF model for the WESAD dataset. From the original 65 HRV features, we selected the HR, HR_SQRT, LF_PCT, MEAN_RR_LOG, MEAN_RR_SQRT, MEDIAN_RR, MEAN_RR, and HF features using the mRMR method. The description of these features is also reported in Table 3. Comparing these features with the features selected from SWELL, shown in the third column of Table 3, one realizes that the selected eight features are data dependent, being different between the SWELL and WESAD datasets. Nonetheless, five selected features are the same: HR, HR_SQRT, MEAN_RR_SQRT, MEAN_RR_LOG, and MEDIAN_RR. After selecting the eight features for the WESAD dataset, we train and test our RF model. Figure 5 and Appendix A report the performance of the trained RF model for each subject of the WESAD dataset. Again, the trained RF model has a classifying power equal to or higher than 99% for each individual subject of the dataset. In particular, except for subjects 4, 5, and 15, the RF model identifies stress events with accuracy, precision, recall, and an F1 score of 100%, validating the global framework to detect stress for the WESAD dataset.

Although the results reported in Figure 4 and Figure 5 validate the global framework for stress detection, these results do not validate the generalization of the stress model. In other words, a generalized global stress model should be able to detect stress conditions regardless of the origin/cause of it. The next validation of the global stress framework examines our capability to classify stress under two causes of stress. Therefore, for this validation, we combined the SWELL and WESAD datasets, having a total of 37 subjects. For this validation, we chose the common features between both datasets to combine them, reducing the number of features from 67 to 63 in the SWELL dataset and 65 to 63 in the WESAD dataset. As the third column of Table 3 shows, the selected mRMR features are dataset dependent; for the combined dataset, the selected eight features were KURT, VLF, MEAN_REL_RR, HR_HF, pNN25, KURT_REL_RR, TP, and MEDIAN_REL_RR_LOG. Again, the description of these features is reported in Table 3. Note that none of these features except for one were previously selected during the individual testing of each dataset. Per previous validations, we use 70% and 30% of each subject’s instances from the combined dataset for training and testing, respectively. Figure 6 and Appendix A report the performance metrics (e.g., accuracy, precision, recall, and F1 score) for each subject of the combined dataset. From these results, we can see the high classifying power of the trained RF model. We can detect stress conditions with a mean accuracy of 99.5% and a standard deviation of 0.8% using KURT, VLF, MEAN_REL_RR, HR_HF, pNN25, KURT_REL_RR, TP, and MEDIAN_REL_RR_LOG as training features. Regarding the other performance metrics, the average and standard deviation of the precision, recall, and F1-score metrics are 99.4% and 0.9% for all the subjects in the combined dataset, enabling the RF model to identify stress events.

## 4. Discussion and Conclusions

Both young and adult people have experienced stress over the last year, becoming one of the main health issues of this decade. For this reason, it is critical to investigate and develop a global stress framework that detects stress events accurately and in a non-invasive way. This work focused on detecting stress levels using HRV features because of the broad acceptance of HRV-based wearable devices used almost daily to track daily events. In this research study, we have investigated the combination of eight HRV features recorded from a single sensor with an RF model to detect stress on an individual subject using a global stress detection framework. The eight HRV features were selected using the mRMR feature selection method [33]. The proposed global stress framework (Figure 1 and Figure 2) has been tested using WESAD and SWELL datasets. The global stress framework is based on the principle that each individual dataset is split into training and testing data in a ratio of 70:30 for each subject. All subjects’ training data were combined and used to select the best eight features using the mRMR feature selection method. An RF model was trained using the best eight features and a stratified 15-fold cross-validation method. Then, the best-trained RF model using the global framework was used to detect stress conditions individually. We have found that the classification power of our trained RF models is equal to or higher than 99% for each subject of both datasets based on conventional evaluation metrics (e.g., accuracy, precision, recall, and F1 score). In addition, we combined both datasets to validate the generalization of the proposed global stress framework. For the combined dataset, we achieved the highest classifier accuracy equal to or higher than 97% for each subject. This result shows that including more subjects in the global training, which introduces more variability in our training dataset, does not reduce the performance of the global stress model on individual testing. This result should be seen as a stepping-stone approach to developing a global stress framework.

The high accuracy of the proposed global stress framework raises some concerns about the possibility of overfitting. Figure 7 compares the accuracy for three training methods using the WESAD dataset. The person-specific approach (e.g., specific training and testing) provides the highest stress prediction (blue line in Figure 7), but this method lacks generalization prediction. The proposed hybrid method with global training and individual testing has similar prediction power to the person-specific model. On the contrary, the accuracy of the leave-one-out approach, in which the training dataset does not include any data related to the tested subject, is highly dependent on the tested subject, ranging from 0.3143 to 0.949. Although these results could be considered the most realistic/generalized model, the individual performance may be deficient for some individuals such as Subject 14, lacking prediction power. However, following the authors in Ref. [22], the prediction model can provide successful results through incorporating person-specific data into the global training dataset. In Ref. [22], the authors discussed a potential calibration model to add these data. Whereas our proposed global stress framework is similar to the one proposed in Ref. [22] in terms of adding person-specific data into the training dataset, the main difference between both works is the number of features used. One of our objectives was to minimize this number of features, reducing the complexity of the model and the training time.

During this research study, we observed that the selected features using the mRMR method depend on the dataset, being different from the three tested cases (i.e., SWELL, WESAD, and combined validations). Additionally, since our model is a random forest (i.e., an ensemble of decision trees), one can use relative feature importance to evaluate the weight of each feature in the trained RF to predict stress. Figure 8 shows the importance scores of the mRMR-selected features for the three datasets. The metric used to measure the feature importance is the popular Gini importance, which calculates the number of instances a feature is used to decide a split in all trees in proportion to the number of samples it had a role in splitting [43]. In other words, the relative importance scores provide a measurement of the utility of each feature in the design of the decision tree. The more a feature is used to make predictions within the decision trees, the greater its relative importance. Whereas five of the eight features are the same in the SWELL and WESAD datasets (HR, HR_SQRT, MEAN_RR_SQRT, MEAN_RR_LOG, and MEDIAN_RR), Figure 8a,b demonstrate that their importance in the trained models are different. This result may be related to the difference between subjects and the cause of stress. Future work should be focused on further analyzing these results. On the other hand, the combined dataset (Figure 8c) has the most unique subset of features (KURT, VLF, MEAN_REL_RR, HR_HF, pNN25, KURT_REL_RR, TP, and MEDIAN_REL_RR_LOG). Nonetheless, some of these features are engineered features. For example, the ‘HR & HF’ feature is highly correlated to the ‘HR’ one. Although the selected eight features of the combined dataset are totally different from the SWELL and WESAD datasets, these eight features also provide a high classifier model when trained using the SWELL and WESAD datasets (see Figure 9). The corresponding tables of these results are Appendix A. Note that the performance of the globally trained RF model is still high; the accuracy of detecting stress events is 100% for each subject in both datasets.

It is important to mention that the idea of global training and individual testing for stress detection was recently published by Fauzi et al. in 2022 [29]. In that study, the authors investigated a stress detection method using 420 features extracted from ST, ACC, EDA, and BVP signals of the WESAD dataset. In Ref. [29], Fauzi et al. investigated three different learning schemes (i.e., individual, centralized, and federated) of a logistic regression (LR) algorithm. Whereas the individual learning approach uses each individual data instance for training and testing, the centralized learning scheme combines all users’ data instances for training but is individually tested. The federated learning scheme operates similarly to the centralized one. The only difference between them is that the federated learning approach maintains the user’s privacy, ensuring that each user’s data never leave the user’s device. Although our proposed global stress framework (Figure 1) and the centralized learning scheme by Fauzi et al. are similar, the performance of the models is quite different. Fauzi et al. reported accuracy, precision, recall, and F1-score values ranging from 0.7475 to 1.00. Using our proposed system, we achieved higher performance matrices ranging from 0.99 to 1.00. Apart from the reported results, the main difference between our work and the recently published work by Fauzi et al. is the input features since they used a multimodal approach, combining features from multiple sensors and training their LR model without prior selection of them. In our work, we have been focused on selecting the top eight features with the highest classifying power to reduce the training time of the global stress platform.

In conclusion, in this study, we have proposed a global stress detection framework combining eight HRV features and an RF model with a classifying power of 99% or higher during individual testing using two benchmark datasets (i.e., WESAD and SWELL datasets). This study presents some limitations related to generalization, lack of diversity, and ability to track the cause of stress. The issues of generalization and lack of diversity are related to the fact that the used datasets contain information from a small number of individuals. Additionally, these measurements were recorded in a controlled laboratory setting, not mimicking a real-world setting or including a larger population. In addition, it is important to acknowledge that the used datasets may not be representative of the general population, since there is no information regarding the individuals’ backgrounds (e.g., age, sex, race). Another potential limitation of this study is the unknown causes of stress, since stress response may differ based on its cause. The trained RF models have been only validated for three types of stress. Future work should be focused on applying the proposed global stress framework in real-world scenarios with a larger and broader population where people experience stress due to different causes. Such study will enable us to investigate the weight of the selected features and their clinical interpretation, aiming to disentangle the relationship between features and causes of stress. In this work, we selected HRV features from a chest sensor since the data are more accurate. Future work should also be focused on validating the proposed global stress framework using HRV features recorded on wristband devices instead of chest-based devices, since their applicability is broader.

## Figures and Tables

**Figure 1 sensors-23-05220-f001:**
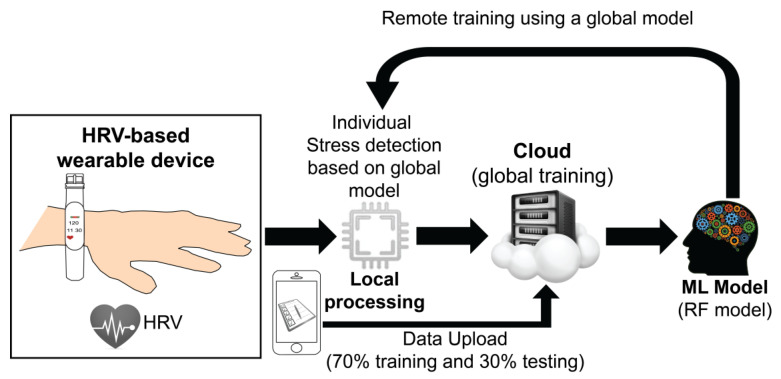
Proposed global stress detection framework.

**Figure 2 sensors-23-05220-f002:**
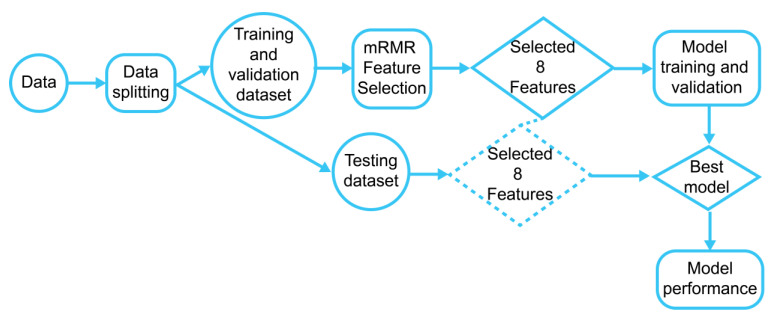
Flowchart of the proposed global stress model using HRV features.

**Figure 3 sensors-23-05220-f003:**
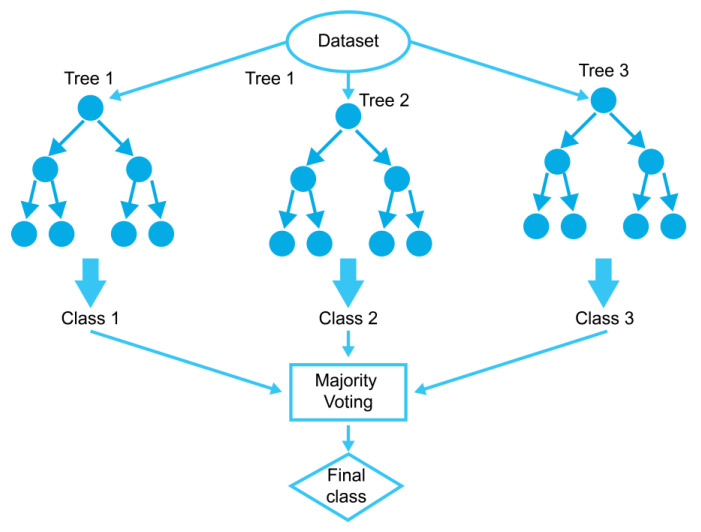
Block diagram of a random forest (RF) classifier model. The output of the RF model is based on the popular vote of the individual decision trees.

**Figure 4 sensors-23-05220-f004:**
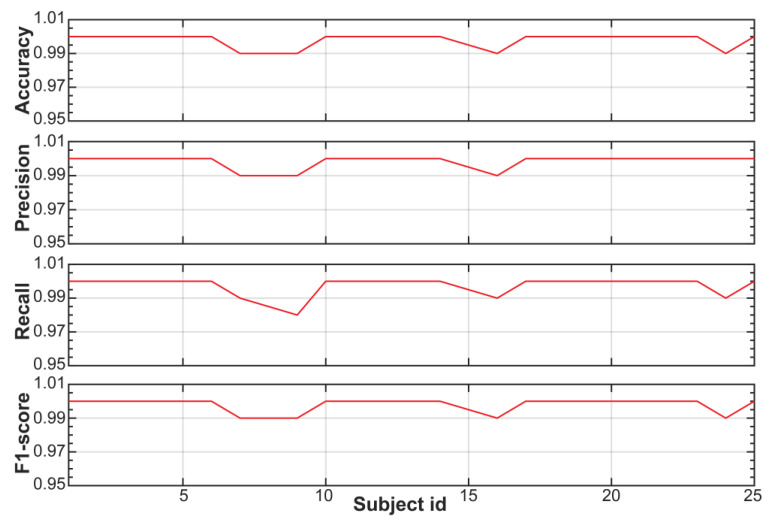
Performance metric of the trained RF model for the SWELL dataset using the global stress framework (e.g., global training but individual testing).

**Figure 5 sensors-23-05220-f005:**
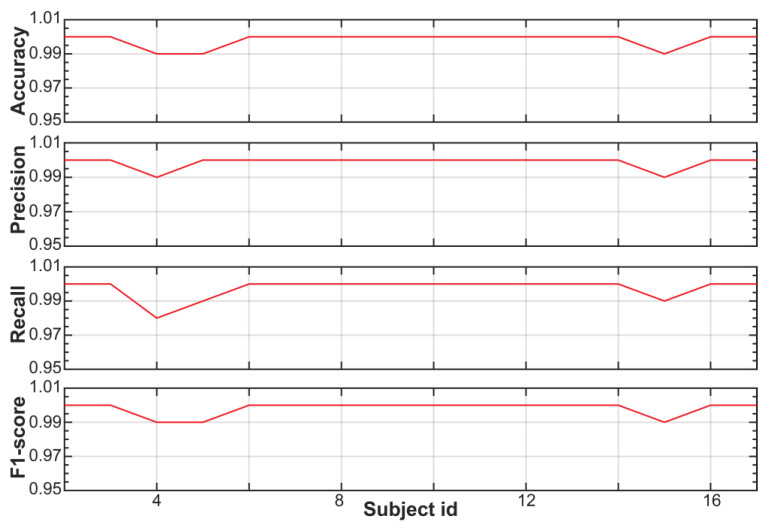
Performance metric of the trained RF model for the WESAD dataset using the global stress framework (e.g., global training but individual testing).

**Figure 6 sensors-23-05220-f006:**
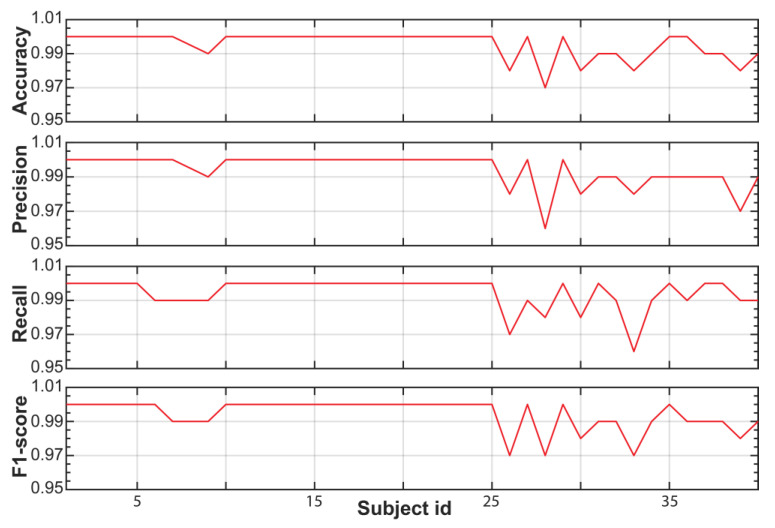
Performance metric of the trained RF model for the combined (SWELL + WESAD) dataset using the global stress framework (e.g., global training but individual testing).

**Figure 7 sensors-23-05220-f007:**
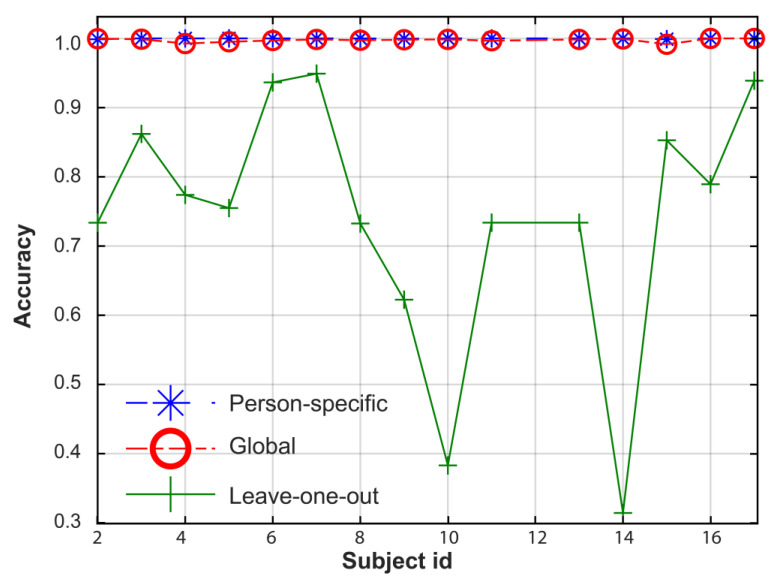
Performance comparison between different training methods tested on individual subjects using the WESAD dataset: (1) person-specific training, (2) global training, and (3) leave-one-out approach.

**Figure 8 sensors-23-05220-f008:**
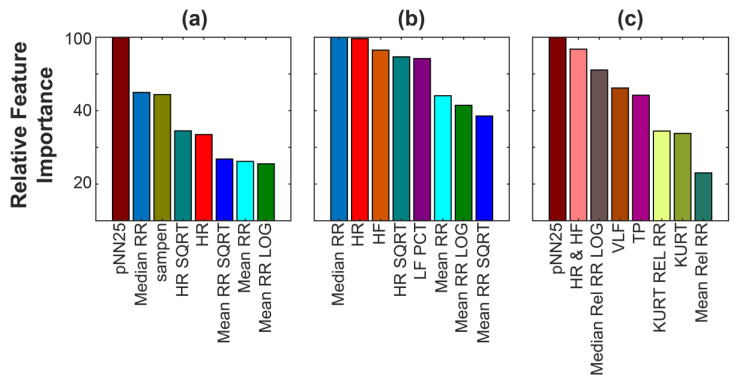
Feature Importance Score for each selected feature in the trained RF models using (**a**) SWELL, (**b**) WESAD, and (**c**) the combined dataset.

**Figure 9 sensors-23-05220-f009:**
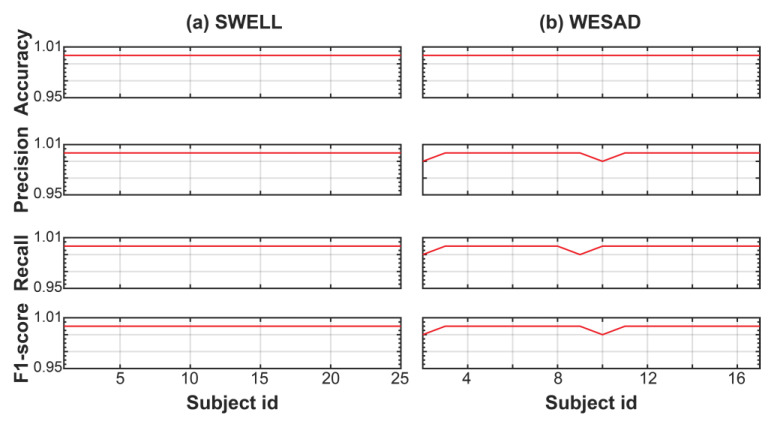
Performance metric of the trained RF model for the SWELL (**a**) and WESAD (**b**) dataset using the global stress framework and the eight selected features from the combined dataset.

**Table 1 sensors-23-05220-t001:** Summary of the datasets used for creating a global stress monitoring device.

Datasets	Subjects	Features	Total Instances Per Feature	Instances vs. Class	Modified Instances vs. Class
WESAD	15	65	135,650	71,640 baseline	94,704 no-stress
				23,064 amusement	
				40,946 stress	40,946 stress
SWELL	22	67	391,638	212,400 no-stress	212,400 no-stress
				110,943 interruption	179,238 stress
				68,295 time pressure	
Combined WEDAD and SWELL	37	63	527,288	220,184 stress	
				307,104 no-stress	

**Table 2 sensors-23-05220-t002:** Confusion matrix.

Confusion Matrix		Actual Values	
		Positive	Negative
Predicted values	Positive	TP	FP
	Negative	FN	TN

**Table 3 sensors-23-05220-t003:** Selected HRV features from SWELL, WESAD, and combined datasets using mRMR method.

Features	Description [16]	Dataset
HR_SQRT	Square root of the mean of the sum of the squared differences between adjacent RR intervals	SWELL WESAD
sampen	Sample entropy	SWELL
MEAN_RR	Mean of all RR intervals	SWELL WESAD
HR	Heart rate (beats per minute)	SWELL WESAD
MEAN_RR_SQRT	Square root of the mean RR interval	SWELL WESAD
MEAN_RR_LOG	Natural logarithm of the mean RR interval	SWELL WESAD
MEDIAN_RR	Median of all RR intervals	SWELL WESAD
LF_PCT	Low-frequency power as a percentage of total power	WESAD
HF	High (0.15 Hz–0.4 Hz) frequency band of the HRV power spectrum	WESAD
pNN25	% of adjacent RR intervals differing by more than 25 ms	SWELL Combined
KURT	Kurtosis of all RR intervals	Combined
VLF	Very low (0.003 Hz–0.04 Hz) frequency band of the HRV power spectrum	Combined
MEAN_REL_RR	Mean of all relative RR intervals	Combined
HR_HF	Heart rate high frequency	Combined
KURT_REL_RR	Kurtosis of all relative RR intervals	Combined
TP	Total HRV power spectrum	Combined
MEDIAN_REL_RR_LOG	Natural logarithm of the median relative RR interval	Combined

## Data Availability

Not applicable.

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
