# Peer review of "Global Stress Detection Framework Combining a Reduced Set of HRV Features and Random Forest Model"

_sensors, 2023, doi:10.3390/s23115220_

Round 1

Reviewer 1 Report

General

This paper aims to develop a stress detection model based on a set of HRV features. The authors utilized two popular datasets SWELL and WESAD to build person-specific models based on reduced HRV features using minimum redundancy maximum relevancy and random forest algorithms. It is an interesting study which attempts to develop a single robust prediction model based on two different datasets. However, I have major concerns. First, the importance of developing person-specific models has not been clearly explained in the text. Second, the issue of validity of the models since the authors achieve almost perfect accuracies 99-100%. I doubt the current version of the manuscript meets the journal’s quality level. Here are my detailed comments which will be useful to the authors

 Abstract

·         The abstract should be rewritten. L 7-14 can be trimmed to for more concise information.

·         Provide more results and implications in brief.

Introduction

·         The authors should provide stronger arguments why they chose to develop person-specific models, rather than generic, considering such models would commonly fail to predict stress among unseen data or people. Besides, these are impractical to deploy these models in real world settings, particularly not for medical-purposes, due to their inflexibility and costly (Nkurikiyeyezu, Yokokubo, & Lopez, 2020)

·         A table summarizing relevant prior ML studies for stress prediction is required to better identify the position of this study to fill the research gaps.

·         It is important to elaborate why the authors selected the minimum redundancy maximum relevance (mRMR) method

Should be Method not Methodology

·         For better readability, a description of the dataset can be presented in a table (150-175) with additional explanation in the text

·         Why the authors chose eight features of HRV should be provided.

·         Please explain how to derive or calculate total instances? Did they represent all features in the dataset?

·         It is also important to provide information with respect

o   how to deal with missing data

o   handle imbalance data

o   the window size and shift or overlapping (in seconds).

·         It is necessary to supply your code and data along with the submission so readers and can better understand and reproduce the work.

·         The authors use maximum depth of 30 which increases the model’s complexity and thus leading to overfitting (Nkurikiyeyezu et al., 2020). Please explain

·         I do not think it is necessary to provide formulas 1-4 and table 3.

Results

·         Some of eight selected features from the combined datasets:  KURT, VLF, MEAN_REL_RR are not widely used in HRV ML literature. Please discuss

Discussion

·         The models achieve very excellent accuracies which raise overfitting issues although indeed person-specific models will have much greater accuracies than generic models. The discussion should deal a lot with the issue, otherwise, the models do not add scientific merits.

·          Limitations of the study including person-specific models should be acknowledged

Presentation

·         The text is a quite difficult to read. It needs subheadings, more tables, or figures for better presentation of the essential information such as description of data sets

Reference

Nkurikiyeyezu, K., Yokokubo, A., & Lopez, G. (2020). Effect of person-specific biometrics in improving generic stress predictive models. Sensors and Materials, 32(2), 703–722. https://doi.org/10.18494/SAM.2020.2650

Author Response

Please read the attached PDF document to find the responses for Reviewer 1.

Reviewer 2 Report

I have read the manuscript. It is very interesting, but it presents some issues that have to be solved. Following, some comments:

                  Format of the paper should be adjusted according with the journal guidelines. Specifically, control the acronyms definition (there are acronyms defined multiple times).

                  Quality of figures should be improved.

                  I suggest introducing the confusion matrix before the performance indices.

                  Did the authors evaluate the statistical difference and the univariate discriminant power of features?

                  The authors should demonstrate the generalization ability of the tool. Thus, I suggest reporting both training and testing results distribution over the K folds.

                  Decision tree are interpretable tool. I suggest reporting the weights and discuss the clinical interpretation of the results.

Author Response

Please read the attached PDF document to find the responses of Reviewer 2

Round 2

Reviewer 1 Report

I am impressed with the thorough revisions the authors have made in response to my concerns. The revised version demonstrates a clear understanding of the issues raised and provides convincing arguments. Their manuscript now has been significantly improved. Their work will make a valuable contribution to the field.

Just a minor suggestion: please enhance the quality of Figure 8.

Keep up good work!

Author Response

Reviewer #1

Comment #0: “I am impressed with the thorough revisions the authors have made in response to my concerns. The revised version demonstrates a clear understanding of the issues raised and provides convincing arguments. Their manuscript now has been significantly improved. Their work will make a valuable contribution to the field.”

Response: We appreciate the reviewer taking the time to carefully review our manuscript and verify that we have addressed all the concerns. We believe that the quality level of the revised manuscript has been improved.

Comment #1: “Just a minor suggestion: please enhance the quality of Figure 8.”

Response: The Figure 8 has been remade and replaced in the revised manuscript.

Comment #2: “Keep up good work!”

Response: Thank you – we will try it!  

Reviewer 2 Report

Authors solved all my comments.

Author Response

Comment #0: “Authors solved all my comments.”

Response: We appreciate the reviewer taking the extra time to verify that we have addressed al his/her previous concerns.